# Effect of Oral Administration with *Lactobacillus plantarum* CAM6 on the Hematological Profile, Relative Weight of Digestive Organs, and Cecal Traits in Growing Pigs

**DOI:** 10.3390/ani13121915

**Published:** 2023-06-08

**Authors:** Cesar Betancur, Yordan Martínez

**Affiliations:** 1Departamento de Ciencias Pecuarias, Facultad de Medicina Veterinaria y Zootecnia, Universidad de Córdoba, Montería 230002, Colombia; cbetancour@correo.unicordoba.edu.co; 2Agricultural Science and Production Department, Zamorano University, Valle de Yeguare, San Antonio de Oriente, Francisco Morazán, Tegucigalpa 11101, Honduras

**Keywords:** probiotics strain, pig, hematology, digestive organ, intestinal histomorphometry, autochthonous microbiota

## Abstract

**Simple Summary:**

Despite scientific evidence and a ban by the European Union, antibiotic growth promoters (AGPs) are still commonly used in intensive pig production systems in many countries to decrease gastrointestinal problems and improve feed efficiency and growth. Concerns over the adverse effects of antibiotics on human health have led to a search for natural alternatives. Previous studies evaluated the effects of *Lactobacillus plantarum* (*L. plantarum*) CAM6, isolated from the Colombian Creole pig Zungo and registered in Genbank, on weaned piglets and sows during gestation–lactation. However, the effect of this probiotic strain on growing pigs is unknown. The results showed that oral administration of *L. plantarum* CAM 6 modified the weight of the digestive organs, the histomorphometry of the concentric layers, and the depth and width of the cecal crypts; decreased cecal lesions; and stimulated the growth of lactic acid bacteria (specifically autochthonous strains) without affecting the blood count compared to the control. These findings suggest that *Lactobacillus plantarum* CAM6 could be considered as an effective alternative to subtherapeutic antibiotics in growing pigs.

**Abstract:**

This study aimed to investigate the effects of oral administration with *L. plantarum* CAM6 on the hematological profile, relative weight of digestive organs, and cecal traits in growing pigs. A total of 36 castrated male pigs [(Landrace × Pietrain) × Duroc] aged 49 to 139 days old were randomly assigned to 3 experimental groups with 12 animals per treatment. The treatments included a control diet without additives (CTRL), a positive control with subtherapeutic antibiotics (TRT1), and CTRL supplemented with 5 mL *Lactobacillus plantarum* CAM6 preparation providing 10^9^ CFU/pig/day (TRT2). The TRT2 group showed a higher (*p* ≤ 0.05) small intestine length and the cecum relative weight compared to the CTRL group. Moreover, *L. plantarum* CAM6 supplementation promoted (*p* ≤ 0.05) increased thickness of the muscular and mucosal layers, as well as enhanced depth and width of the cecal crypts. The TRT2 group also showed well-defined crypts without lesions, while the CTRL and TRT1 groups exhibited congestion, lymphocytic infiltration in the crypt, and intestinal-associated lymphoid tissue atrophy, respectively. Additionally, TRT2 stimulated (*p* ≤ 0.05) the growth of the autochthonous cecal microbiota compared to other experimental groups. Overall, the results indicate that oral administration of *L. plantarum* CAM6 improved intestinal health and enhanced the growth of autochthonous cecal lactic acid bacteria and had no impact on the complete blood count in growing pigs.

## 1. Introduction

The increased demand for safe and high-quality pork requires the use of diets containing natural and safe feed supplements that positively impact animal health and pork quality [1]. With the adoption of advanced technology guaranteeing high levels of productivity in intensive pig production systems [2], animals are inadvertently exposed to various stressors [3], which predisposes them to digestive diseases that lead to a decrease in growth performance and production [4]. Despite the ban on antibiotic growth promoters (AGPs) in the European Union, they are still commonly used in intensive pig production systems in many countries. However, concerns over the adverse effects of antibiotics on microbial resistance, cross-resistance with other microorganisms, and bioaccumulation in muscles have led researchers to explore the use of natural alternatives, including probiotics, prebiotics, symbiotics, postbiotics, enzymes, lauric acid, and antimicrobial peptides [2,3]. 

In this sense, probiotics have shown promise in counteracting the negative effects of stress and improving gastrointestinal health in pigs [5], although the use of probiotics in growing pigs has been relatively limited due to the maturity of the gastrointestinal tract, with highly enzymatic digestive activity, immunological capacity, and disease resistance. However, there is evidence suggesting that alterations in the gastrointestinal barrier function observed in weaned pigs persist in adulthood [6]. Consequently, oral supplementation with probiotics has the potential to positively impact the health and productivity of pigs. Numerous studies have demonstrated that probiotic use enhances intestinal histomorphometry, nutrient absorption, immune system function, antioxidant capacity, and intestinal permeability, as well as promoting intestinal eubiosis [7,8]. Among the various bacteria used as probiotics, the genus *Lactobacillus*, specifically colonizing lactic acid bacteria of the gastrointestinal tract, stands out as the most commonly used probiotic agent [9]. Notably, the probiotic strain *Lactobacillus plantarum* has exhibited gastrointestinal health benefits in pigs [10,11]. Previous findings indicate that oral administration of *Lactobacillus plantarum* CAM6 improves growth performance, carcass traits, and humoral immunity, while also reducing the serum glucose, cholesterol, and beta-hydroxybutyrate concentrations in weanling piglets [12]. Additionally, this specific probiotic strain (CAM6) enhances the chemical composition of sow milk, consequently improving the productivity and health of their offspring [13]. 

Despite several studies demonstrating the beneficial effects of *Lactobacillus* spp. strains, only a limited number of studies have investigated the effect of *Lactobacillus plantarum* strains isolated from Creole pigs on biological indicators of pigs. Hence, our hypothesis is that oral supplementation with *L. plantarum* CAM6 could stimulate the population of lactic acid bacteria, and consequently improve intestinal integrity while reducing cecal damage without affecting the health indicators of the blood. This experiment aims to evaluate the effects of oral supplementation with *L. plantarum* CAM6 on the hematological profile, relative weight of digestive organs, and cecal microbiology in growing pigs.

## 2. Materials and Methods 

### 2.1. Ethics Statement

The experiment was conducted according to the guidelines stablished by the National Institute of Animal Health, and all animal procedures were approved by the Research Ethics Committee of the University of Córdoba, Colombia (Statement No. 001 of 26 January 2020).

### 2.2. Probiotic Preparation, Animals, and Experimental Treatments

The *L. plantarum* CAM6 strain used in this study was isolated from the gastrointestinal tract of Creole hairless pigs (Zungo) found on the north coast of Colombia [14]. The bacterial strain (CAM6) was inoculated in a mixture of pineapple, banana, and papaya peel juice, which provided a similar growth environment to the MRS culture medium [15]. The bacterial strain was registered in GenBank with access number MK523644.1 (https://www.ncbi.nlm.nih.gov/nuccore/1573758920?log$=activity), accessed on 1 August 2022.

A total of 36 male (castrated) pigs, crossbred [(Landrace × Pietrain) × Duroc] with an initial body weight of 12.5 ± 0.18 kg were selected from 20 sows at the same time. The pigs were randomly assigned to 3 experimental groups, with 12 animals per group and 4 animals per pen. Each pen served as an experimental unit for the duration of the study, which spanned from 49 to 139 days of age. The experimental treatments consisted of (1) a control diet without additives (CTRL), (2) CTRL supplemented with 350 mg/kg colistin sulfate (20% active compounds) (Ceva Santé Animale, Libourne, France) as a growth promoter (TRT1), and (3) CTRL plus *L. plantarum* CAM6 at a ratio of 5 × 10^9^ CFU/mL (TRT2). The dosage of the probiotic used was determined based on the results of previous studies in pigs, including weaners and breeders [12,13,14]. The basal diet was formulated to meet the nutrient requirements for pigs with a body weight of 10 to 50 kg [16] (Table 1).

### 2.3. Housing and Experimental Conditions

The experiment was conducted at the swine research laboratory of the University of Córdoba, Berastegui campus, Córdoba, Colombia. Each pen, measuring 4 × 2 × 0.6 m, housed four pigs. The pens were constructed with concrete floors and precast concrete walls, equipped with canoe linear feeders and nipple-type drinkers. The pigs were kept under natural ventilation and subjected to a lighting regime. The average humidity and temperature in the facility were maintained at 55% and 26 ± 1 °C, respectively.

The pigs were dewormed and vaccinated for swine fever after 28 days of weaning, and no drugs were administered during the experimental period. The antibiotic was mixed with the feed and the probiotic suspension was orally administered using a syringe at 7:00 a.m. throughout the entire experimental period. Feed was provided *ad libitum* twice a day (at 8:00 a.m. and 3:00 p.m.), and the pigs had free access to water for the 90-day duration of the study. 

### 2.4. Blood Profiles

At the conclusion of the experiment, 12 pigs were randomly selected from each treatment group after fasting, and blood samples of 5 mL were collected through jugular vein puncture into heparinized Vacutainer tubes with lilac caps (BD Vacutainer; BD, Franklin Lakes, NJ, USA). The tubes, placed in containers with ice packs, were transported to the laboratory and kept at 4 °C until centrifugation at 1381× *g* (Eppendorf centrifuge AG, New York, NY, USA) for 15 min at room temperature.

The plasma samples were processed at the Veterinary Clinical Laboratory of the Faculty of Veterinary Medicine and Zootechnics, University of Córdoba. The samples were analyzed for white blood cells (WBC), red blood cells (RBC), hemoglobin, hematocrit, mean corpuscular volume (MCV), mean corpuscular hemoglobin concentration (MCHC), mean cell volume (MCV), and mean platelet volume (MPV) using a semiautomatic analyzer (Horiba ABX Micros ESV 60^®^; Paris, France).

### 2.5. Intestinal Tissue Sampling and Histomorphology Measurement

At the end of the 90-day experimental period, when the pigs reached a final weight of 56.5 ± 0.18 kg, two pigs were randomly selected from each pen, resulting in a total of six pigs per treatment group. The pigs were sedated with an intramuscular injection of 5 mL of Xylazine (Rompum^®^, Bayer Health Care, Animal Health Division, KS, USA) before slaughter, then euthanized by exsanguination through the jugular vein. The pigs were positioned in the dorsal decubitus position, and a midline laparotomy was performed. An incision was made in the abdomen from the sternum to the pubis, exposing the entire gastrointestinal tract (GIT). The intestines were freed from the mesentery, and the length of the small intestine was measured using a tape measure (Stanley Hand Tools 33–42, CT, USA). The relative weights of the small intestine, the large intestine, and the cecum were determined by considering the body weight at euthanasia and the weight of each empty organ.

The histomorphology measurements were made with tissue samples measuring 5 to 8 cm in length from the cecum of each animal and immediately fixed in 10% neutral-buffered formalin for 24 h, then dehydrated using increasing concentrations of ethanol (70% to 90%), followed by absolute ethanol, isopropyl alcohol, and xylol. Dehydrated segments were embedded in paraffin and refrigerated to achieve sufficient paraffin hardness. Cross sections of each sample, approximately 4 µm thick, were stained with hematoxylin and eosin and examined by a veterinary pathologist using a light microscope with a microtome (Leica MZ16A, Bensheim, Germany). The histomorphological examination involved triplicate measurements of the thickness of the muscle and mucosa layer (six measurements per treatment), as well as the depth and width of the crypts. The histological measurements were performed at the Animal Pathology Laboratory of the Diagnostic Center of CORPOICA Montería, Colombia.

### 2.6. Cecal Microbiology

To assess the cecal microbiology, the mucosal layer of each cecal sample (n = 6 per treatment) was scraped using a scalpel to obtain digesta for microbial culturing. For each sample, 1 g of mucosal layer was added to a tube containing 9 mL of sterile buffered peptone water. The mixture was homogenized in distilled water at a ratio of 1/10 (*w*/*w*), and serial dilutions ranging from 10^6^ to 10^9^ were prepared. From each dilution, 0.1 mL was taken in triplicates and plated on Man Rogosa and Sharpe (MRS) agar (Difco Laboratories, Detroit, MI, USA). The plates were then incubated at 37 °C and pH 5.6 for 48 h under anaerobic conditions using the Gas Pak system (BBL, Rockville, MD, USA). The colony counts were expressed as log^10^ CFU/g. Lactic acid bacteria were identified using an API^®^ 50 CHL gallery. For molecular identification of cecal lactic acid bacteria, the 1465 bp region of the 16S ribosomal gene was amplified through polymerase chain reaction (PCR) using the universal primers F27 (50-AGAGTTTGAT CMTGGCTCAG-30) and R1492 (50-TACGGYTACCTTGTTACGACTT-30). The amplified fragments were purified and sent to a reference laboratory for sequencing (Macrogen Inc., 2017, Seoul, Republic of Korea). The sequenced products were evaluated using the basic local alignment search tool. 

Microbiological analysis was performed using a Leica microsystem (LAs EZ microscope, Bensheim, Germany) connected to a Leica DM 500 computer. Images were obtained at 100× and 500× magnification, and measurements were made using an Axion Vision 4.8 image analyzer. These microbiological determinations were conducted at the biotechnology laboratory (GRUBIODEQ), University of Córdoba, Colombia.

### 2.7. Statistical Analysis

All data obtained were subjected to analysis of variance (ANOVA) under a completely randomized design. Prior to the analysis, normality and homogeneity of variance were tested using the Kolmogorov–Smirnov and Bartlett tests, respectively. If necessary, Duncan’s multiple range test was performed to separate the means. The statistical analyzes were performed using SPSS version 21 [17], and *p*-value less than 0.05 was considered significant.

## 3. Results

### 3.1. Hematological Parameters

The impact of *L. plantarum* CAM6 on the hematological parameters of growing pigs is summarized in Table 2. Pigs in the TRT1 group exhibited higher (*p* ≤ 0.05) concentrations of leukocytes, lymphocytes, and granulocytes and lower (*p* ≤ 0.05) concentrations of hemoglobin compared to the CTRL and TRT2 groups (except for granulocytes). The other hematological indicators did not show significant changes (*p* > 0.05) due to the effect of the experimental groups. 

### 3.2. Relative Weight of the Digestive Organs

The relative weight of the small intestine and large intestine did not affect by the treatments (*p* > 0.05) during the experimental period as shown in Table 3. However, pigs in the TRT2 group showed higher (*p* ≤ 0.05) relative weight of the cecum compared to the CTRL and TRT2 groups. Additionally, the length of the small intestine was higher in TRT1 and TRT2 groups compared to the CTRL group.

### 3.3. Cecal Histomorphology 

Table 4 presents the effect of oral administration with *Lactobacillus plantarum* CAM6 on cecal integrity as an indicator of gut health. TRT2 significantly improved (*p* ≤ 0.05) the mucous and muscle layer thickness, as well as cecal crypt depth and width in growing pigs compared to the other treatments. Conversely, the control diet without additives showed the smallest muscle layer thickness (*p* ≤ 0.05) and the lowest values for the cecal crypts.

The control group caused congestion (Figure 1a) and crypt atrophy (Figure 1b). Furthermore, the TRT1 group was liable to cause edema (Figure 1c) and GALT (gut-associated lymphoid tissue) with lymphocytic infiltration (Figure 1d). The TRT2 group indicated well-defined and lesion-free crypts (Figure 1e). 

### 3.4. Cecal Microbiology 

Figure 2 displays the effect of oral administration of *Lactobacillus plantarum* CAM6 on the growth of cecal lactic acid bacteria in growing pigs. Supplementation with the CAM6 strain significantly stimulated (*p* ≤ 0.05) the growth of cecal lactic acid bacteria in comparison to the CTRL diet, but showed comparable results with the subtherapeutic antibiotic (TRT1) (*p* > 0.05). Furthermore, the oral administration of the CAM6 strain led to the molecular identification of *Lactobacillus reuteri* DSM 20016 (Figure 3) and *Lactobacillus johnsonii* CIP 103620 (Figure 4) in the cecum of growing pigs. However, *Lactobacillus plantarum* CAM6 was not detected in this intestinal portion.

## 4. Discussion

The use of probiotics or beneficial microorganisms in the swine industry has gained importance in recent years as an alternative to growth-promoting antibiotics and to improve intestinal health in pigs [19]. In this study, oral supplementation with *L.s plantarum* CAM6 (as a probiotic strain) resulted in improved feed efficiency and reduced diarrhea in pigs compared to the control diet. Hematological parameters are commonly used as indicators of health in animals without apparent diseases and symptoms. Bacterial, viral, parasitic, or fungal infections can be diagnosed by variations in blood count [20].

The oral administration of colistin has been shown to affect white blood cell concentration in growing pigs. However, the effects were not observed in weaned pigs [12]. Certain synthetic or natural compounds with antimicrobial properties can induce changes in polymorphonuclear leukocytes, primarily by activating the immune system to eliminate exogenous material or potential toxic and allergenic compounds. Elevated levels of these cellular elements in the blood may indicate the presence of infections [21]. It is worth noting that the hematological parameters observed in this study were within the normal ranges for the species [18].

Oral antibiotics are known to promote the maturation of systemic immunity and delay intestinal bacterial colonization [22]. This suggests that oral administration of antibiotics accelerates blood neutrophil maturation, which may affect other blood parameters. In this study, TRT1 pigs had higher leukocyte and lymphocyte counts. Foster et al. [23] reported that oral administration of probiotic strains increased blood polymorphonuclear neutrophils and villi height in young pigs. Huang et al. [24], demonstrated increased immune activation and concentration of polymorphonuclear leukocytes in apparently healthy weanling pigs treated with various nutraceutical-related products. However, other studies using subtherapeutic antibiotics or functional products did not observe changes in the concentration of polymorphonuclear leukocytes in the blood of pigs [25,26]. The changes in serum polymorphonuclear cell concentrations within normal parameters may depend on various factors such as pig genetics, production conditions, the type and duration of antibiotics, or natural product usage, and further research is needed to confirm this hypothesis.

Furthermore, the use of preventive antibiotics (TRT1) resulted in a decrease in hemoglobin concentration. Although there are scientific contradictions regarding whether certain antibiotics used in pigs can cause hemolytic anemia [27], our study found that the prolonged use (90 days) of colistin resulted in a decrease in hemoglobin levels by 1.76 g/dL. On the other hand, antibiotic growth promoters are known to have anti-inflammatory and antimicrobial effects, benefiting nutrient absorption [28]. However, prolonged antibiotic use can disrupt the intestinal microflora and the nutrient homeostasis [29]. In this sense, Méhi et al. [30] reported that long-term antibiotic use and microbial resistance can disrupt iron homeostasis due to the inactivation of a central transcriptional regulator. Apparently, this nonpathological effect may cause a decrease in hemoglobin levels in growing pigs.

Interestingly, the length of the small intestine was higher in the TRT1 and TRT2 groups, which may be associated with changes in the intestinal microflora and improved gut health, resulting in a larger surface area for nutrient absorption [31]. A decrease in the concentration of metabolites or toxins can influence intestinal morphology and increase epithelial cell proliferation [32]. Moreover, Hou et al. [33] reported that *Lactobacillus reuteri* can modify the morphology and motility of the small intestine and reduce the growth of opportunistic pathogens in pigs. However, Cilieborg et al. [34] found that the use of *Lactobacillus paracasei*, *Bifidobacteria animalis*, and *Streptococcus thermophilus* decreased the relative weight of the small intestines of pigs with necrotizing enterocolitis. Additionally, Matuer and Eraslan [35] reported that oral probiotics can increase the absorption surface area in the small intestine, particularly in the jejunum. The effects of probiotics on intestinal morphology may depend on the colonization ability of the specific microbial strain used and the health status of the pigs.

Furthermore, the oral administration of *L. plantarum* CAM6 increased the relative weight of the cecum, which can be attributed to the diet and the metabolic processes of the resident microbes [36]. Probiotics, especially those from *Lactobacillus* spp., can colonize the cecal epithelium through the fermentation of complex sugar molecules, leading to the production of short-chain fatty acids that stimulate the proliferation of the intestinal epithelium. This, in turn, influences the relative weight of the cecum and length of the small intestine [37,38]. In this sense, Ayala et al. [32] found that the use of a commercial probiotic increased the relative weight of the cecum in growing Yorkland × CC21 pigs, which was attributed to competitive exclusion and enhanced cecal fermentation.

Changes in intestinal morphology have been associated with improved gut health and productivity in pigs [32]. Previous studies [39,40] have shown that increasing the depth and thickness of cecal crypts enhances the absorption of electrolytes and water in pigs. In this study, the oral administration of *L. plantarum* CAM6 resulted in improved integrity of the muscular and mucosal layers, as well as increased depth and width of the cecal crypts in growing pigs. Liu et al. [41] reported that oral administration of *L. plantarum* 23-1 promoted competitive exclusion and increased mucosal layer thickness, which improved intestinal permeability, gene expression of binding proteins, and nutrient absorption. Additionally, Giang et al. [42] demonstrated that thicker mucosal and muscular layers reduce the adherence of pathogenic bacteria and promote the colonization of beneficial microorganisms in the intestinal lumen. Studies by Yoshida et al. [43] and Wang et al. [44] have reported that *Lactobacillus plantarum* Lq80 and *Lactobacillus plantarum* PFM 105 enhanced intestinal barrier function and productivity in pigs, respectively.

Figure 1 a–e demonstrates that the utilization of the *L. plantarum* CAM6 isolated from the large intestine of Creole pigs, resulted in a well-defined cecum and colonization of lactic acid bacteria in the cecum. Perez et al. [45] reported that increased migration of specialized cells towards the large intestine promotes a reduction in the adhesion of pathogenic bacteria between the crypts, thereby benefiting the colonization and diversity of the intestinal bacteria population. Furthermore, the group receiving subtherapeutic antibiotics provoked GALT in the cecum of the pigs (Figure 1d), which appeared to be associated with an increase in serum lymphocyte concentration (Table 2). GALT, which is a component of the immune system capable of distinguishing antigens and pathogens, is more commonly found in the ileum, and is related to lymphocyte concentration in lymphatic circulation [46]. However, Pérez-Bosque [47] noted that uncontrolled immune responses during GALT activation may result in tissue damage. Likewise, Ruth and Field [48] demonstrated that GALT induces cellular energy expenditure. Therefore, the use of amino acid mixtures may optimize the immune function in both healthy and sick animals.

The importance of the intestinal microbiota in maintaining intestinal homeostasis is well known [49]. The results indicated that oral administration of *L. plantarum* CAM6 had an impact on the native microbial community, leading to an increased population of *Lactobacillus* spp. in the intestine of adult pigs compared to the control and antibiotic groups (Figure 2). Vigors et al. [50] also reported a strong correlation between feed efficiency and the population of *Lactobacillus* spp. in the pig cecum. Moreover, Wang et al. [51] observed that supplementation with *L. plantarum* in combination with FOS increased the LAB population in the cecum compared to animals receiving a control diet and antibiotics. Similarly, Vera et al. [52], reported a significant increase in the population of *Lactobacillus* spp. in the cecal content of pigs due to the administration of *L. plantarum* 22 CML.

According to Peng et al. [53], dietary probiotics have a positive impact on modifying the abundance and diversity of beneficial bacteria in the gastrointestinal tract. The study found that oral supplementation of *L. plantarum* CAM6 altered the diversity of the intestinal microbiota of pigs. Although *Lactobacillus reuteri* DSM 20016 (Figure 3) and *L. johnsonii* CIP 103620 (Figure 4) were identified, *L. plantarum* (CAM6) was not identified in the cecum of growing pigs. Similarly, Suo et al. [54] demonstrated that *L. plantarum* was not the predominant lactobacillus species in pigs, and Lähteinen et al. [55] discovered that only one isolate among several lactobacillus species recovered from fecal bacterial cultures had a genotype similar to the orally administered strain, called *L. reuteri* GRL 1170. Additionally, Takahashi et al. [56] found that oral administration of *L. plantarum* Lq80 increased the diversity of *lactobacillus*, where *L. reuteri* and *L. crispatus* were the dominant species within the lactobacillus population. These findings suggest that the growth of indigenous *lactobacillus* is promoted by *L. plantarum* CAM6, indicating that this strain has the ability to modulate the intestinal flora and produce molecules that stimulate the growth of autochthonous intestinal lactobacilli in pigs.

## 5. Conclusions

The oral administration with *L. plantarum* CAM6 isolated from the cecum of Creole pigs resulted in modifications to the length of the small intestine and the relative weight of the cecum in growing pigs. It also increased the histomorphometry of the concentric layers (muscle and mucosa) and the depth and width of the cecal crypts. Hematological parameters remained within normal ranges for the studied animal species, although the antibiotic group exhibited increased polymorphonuclear and lymphocyte counts and a decreased hemoglobin concentration. Furthermore, this probiotic strain (CAM6) stimulated the growth of certain indigenous lactobacilli, although further investigation is required to understand this microbial effect better. It is recommended to conduct a scale-up test using the CAM6 strain of *L. plantarum* in drinking water and/or feed to determine the biological response of growing pigs.

## Figures and Tables

**Figure 1 animals-13-01915-f001:**
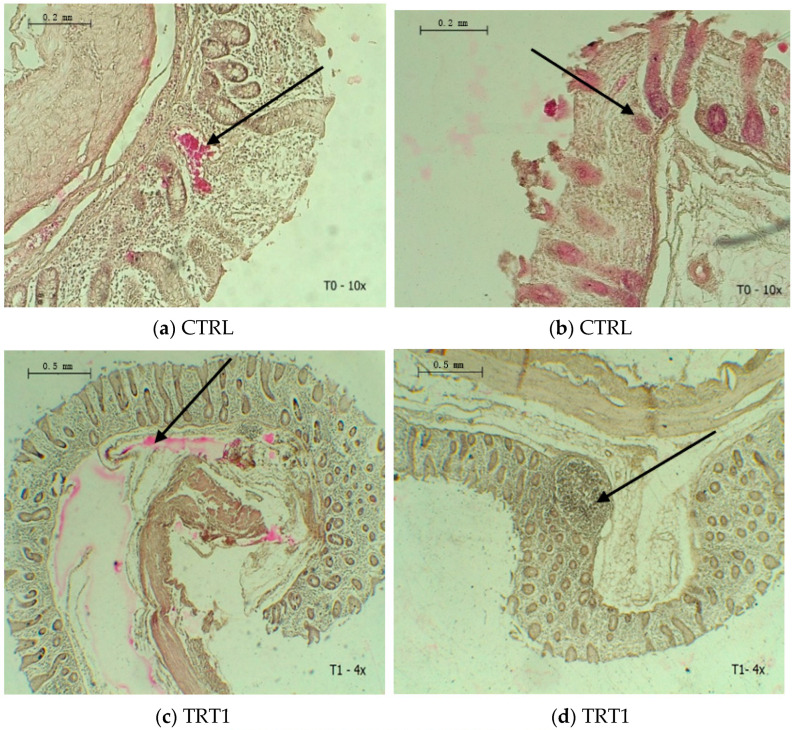
(**a**–**e**). Microscopic images illustrate the effect of dietary treatments supplemented with *L. plantarum* CAM6 on the histopathological analysis of the cecum. CTRL: control group; TRT1: antibiotic group; TRT2: biopreparation containing 5 × 10^9^ of *L. plantarum* CAM6.

**Figure 2 animals-13-01915-f002:**
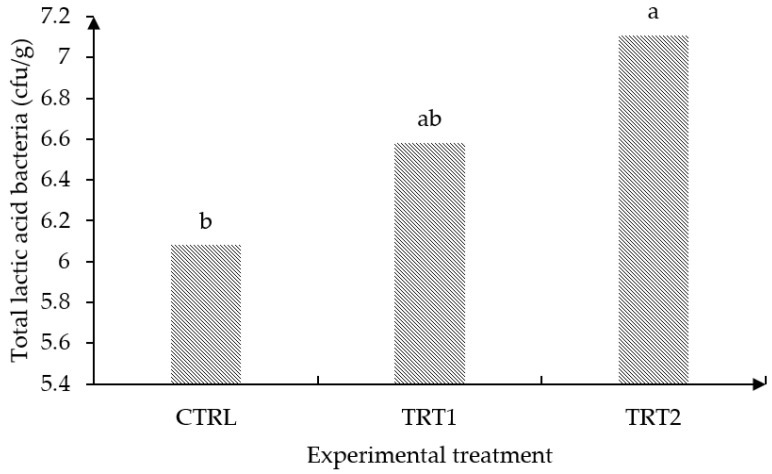
Effect of oral administration with *Lactobacillus plantarum* CAM6 on the growth of cecal lactic acid bacteria in growing pigs (SEM ± 0.181; *p* value = 0.015; n = 18). Means with different letters on top of the bars differ in *p* ≤ 0.05. CTRL: control group; TRT1: antibiotic group; TRT2: biopreparation containing 5 × 10^9^ of *Lactobacillus plantarum* CAM6.

**Figure 3 animals-13-01915-f003:**
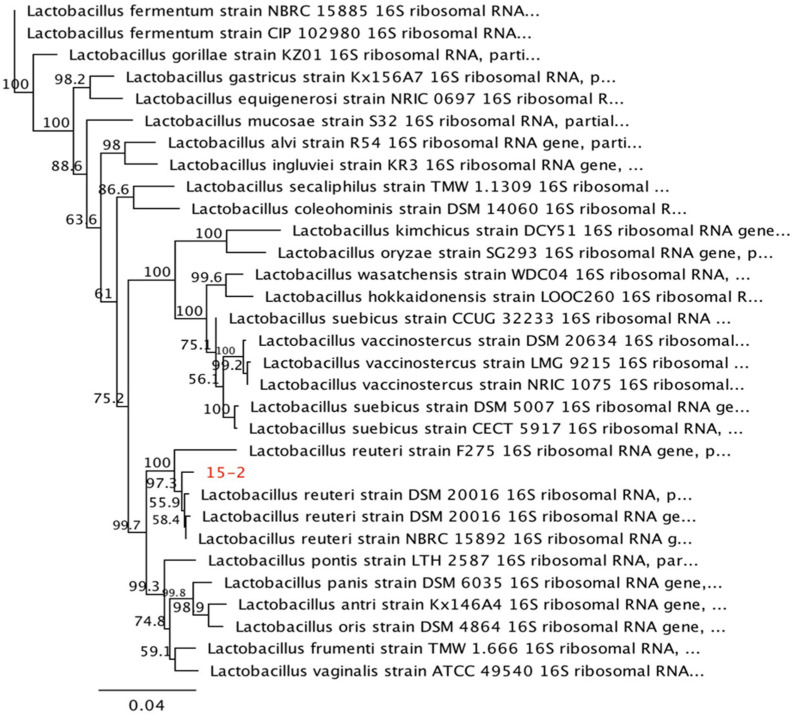
Molecular identification of *Lactobacillus reuteri* DSM 20016 isolated from the cecum of growing pigs who received oral administration with a biopreparation containing 5 × 10^9^ of *Lactobacillus plantarum* CAM6.

**Figure 4 animals-13-01915-f004:**
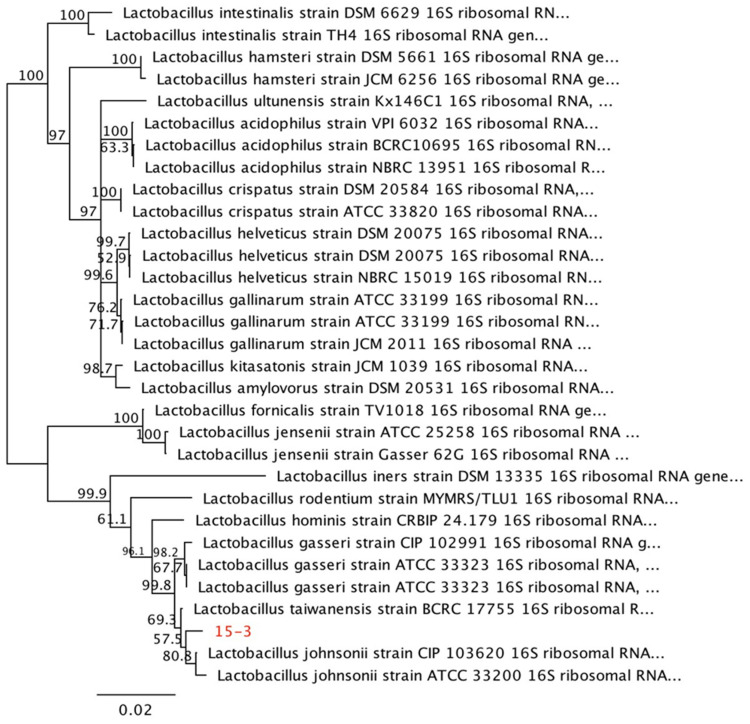
Molecular identification of *Lactobacillus johnsonii* CIP 103620 isolated from the cecum of growing pigs who received oral administration with a biopreparation containing 5 × 10^9^ of *Lactobacillus plantarum* CAM6.

**Table 1 animals-13-01915-t001:** Ingredients and nutritional contributions of the pig diet (49–139 days).

Ingredients	%
Cornmeal	41.24
Full-fat soybean meal	16.00
Wheat bran	18.29
Defatted soybean meal	13.47
Sorghum	7.00
Premix ^1^	4.00
Nutritional contributions	
Crude protein, %	18.0
Calcium, %	1.13
Phosphorus total, %	0.95
Metabolizable energy, MJ/kg	13.39

^1^ The vitamin–mineral premix provided per kilogram of diet: 20,000 IU of vitamin A; 4000 IU of vitamin D_3_; 80 IU of vitamin E; 16 mg of vitamin K; 4 mg of thiamine; 20 mg of riboflavin; 6 mg of pyridoxine; 0.08 mg of vitamin B_12_; 120 mg of niacin; 50 mg of Ca-pantothenate; 2 mg of folic acid; 0.08 mg of biotin; 15 mg of Cu (as copper sulfate); 56 mg of Zn (as zinc oxide); 73 mg of Mn (as manganese oxide); 0.3 mg of I (as potassium iodate); 0.5 mg of Co; 0.4 mg of Se.

**Table 2 animals-13-01915-t002:** Effect of oral administration with *Lactobacillus plantarum* CAM6 on the hematological parameters in growing pigs (n = 36).

	Experimental Treatments					
Items	CTRL	TRT1	TRT2	SEM±	*p* Value	Normal Parameters [18]
Leukocytes, 10^9^/L	14.36 ^b^	17.78 ^a^	15.17 ^b^	1.1008	0.037	11.6–32.9
Lymphocytes, 10^9^/L	4.49 ^b^	5.58 ^a^	4.67 ^b^	0.264	0.019	3.6–18.5
Monocytes, 10^9^/L	1.27	1.53	1.25	0.150	0.587	0.0–4.9
Granulocytes, 10^9^/L	8.60 ^b^	10.66 ^a^	9.25 ^ab^	0.651	0.012	0.3–15.2
Eosinophils, 10^9^/L	0.38	0.46	0.38	0.102	0.086	0.0–2.5
Erythrocytes, 10^12^/L	8.99	8.08	8.56	0.369	0.060	5.7–8.3
Hb, g/dL	13.18 ^a^	11.42 ^b^	13.09 ^a^	0.495	0.050	10–15
Hto, %	45.27	40.66	44.73	1.593	0.133	29–46
MCV, f/L	50.25	50.13	52.25	0.799	0.092	44–56
MCH, pg	14.70	14.16	15.25	0.608	0.447	15–20
MCHC, g/dL	29.16	28.07	29.21	0.511	0.657	32–38
Platelets, 10^9^/L	798.75	742.38	656.38	50.107	0.507	171–833
MPV, f/L	14.79	14.86	13.70	0.417	0.183	7.2–15.6

^a,b^ Means with different letters in the same row differ from *p* ≤ 0.05. CTRL: control group; TRT1: antibiotic group; TRT2: biopreparation containing 5 × 10^9^ of *Lactobacillus plantarum* CAM6. Hb: hemoglobin; Hto: hematocrit; MCV: mean cell volume; MCH: mean corpuscular hemoglobin; MCHC: mean corpuscular hemoglobin concentration, MPV: mean platelet volume.

**Table 3 animals-13-01915-t003:** Effect of oral administration with *Lactobacillus plantarum* CAM6 on intestines (weight and length) traits in growing pigs (n = 18).

	Treatments				
Indicators	CTRL	TRT1	TRT2	SEM±	*p* Value
Small intestine, g/100 g	2.51	2.80	2.55	0.192	0.181
Large intestine, g/100 g	3.82	4.10	3.95	0.115	0.247
Cecum, g/100 g	0.28 ^b^	0.27 ^b^	0.35 ^a^	0.018	0.026
Small intestine, cm	16.51 ^b^	17.58 ^a^	17.24 ^a^	0.262	0.045

^a,b^ Means with different letters in the same row differ from *p* ≤ 0.05. CTRL: control group; TRT1: antibiotic group; TRT2: biopreparation containing 5 × 10^9^ of *Lactobacillus plantarum* CAM6.

**Table 4 animals-13-01915-t004:** Effect of oral administration with *Lactobacillus plantarum* CAM6 on the cecal integrity of growing pigs (n = 18).

	Treatments				
Items	CTRL	TRT1	TRT2	SEM±	*p* Value
Mucous thickness, μm	36.17 ^b^	37.75 ^b^	45.08 ^a^	0.414	˂0.001
Muscle thickness, μm	17.67 ^c^	25.67 ^b^	30.33 ^a^	0.417	˂0.001
Crypts depth, μm	20.08 ^c^	26.92 ^b^	36.25 ^a^	0.811	˂0.001
Crypts width, μm	7.00 ^c^	8.75 ^b^	10.42 ^a^	0.278	˂0.001

^a–c^ Means with different letters in the same row differ from *p* ≤ 0.05. CTRL: control group; TRT1: antibiotic group; TRT2: biopreparation containing 5 × 10^9^ of *Lactobacillus plantarum* CAM6.

## Data Availability

The data presented in this study are available on request from the corresponding author.

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
