# Peer review of "Effect of Oral Administration with *Lactobacillus plantarum* CAM6 on the Hematological Profile, Relative Weight of Digestive Organs, and Cecal Traits in Growing Pigs"

_animals, 2023, doi:10.3390/ani13121915_

Round 1

Reviewer 1 Report

This paper evaluated the effects of Lactobacillus plantarum CAM6 supplementation on the hematological profile, relative weight of digestive organs, and cecal traits in growing pigs. The paper is well-organized, clearly presented, and uses appropriate methodologies. The results are interesting and within the scope of the journal. However, several issues that are necessary to improve the quality of the paper before considering for acceptance.

Specific comments

It is noted that the paper needs careful editing by someone with expertise in technical editing paying particular attention to English grammar, technical expressions, spelling, and sentence structure.

L29: The TRT2 group..

L 30: Specify the name of the probiotic strain.

L 32: the CTRL and TRT1 groups showed..

L 43: By adopting..

L 50: use a comma after the word “thus”.

L 53-55: “feeding pigs diets supplemented” change to “diets supplemented”.

Hypothesis of this work should be included end of the introduction.

L 83: Word capitalization should be consistent throughout the manuscript.

L 85: Abbreviation should be defined as its first appearance in the text; then, use abbreviation throughout the text.

L 93-96: What is the basis for dose selection? Please explain this and clearly mention it in the M&M.

Table 1: What is the difference between Soybean meal and Soymeal?

Table 1: Change “Grow Nucleus” to “Premix”

Table 1: Convert the unit “Kcal/kg” to “MJ/kg”

L 101-102: D3>> D3, B12>> B12

L 107-116: More details of housing conditions; e.g., temperature, humidity, ventilation, etc.

L 107-116: How about the growth performance and feed intake/refusal of piglets?

L 123: rpm should be converted into × g,

L 123: Generally, centrifugation to obtain plasma should be conducted at 4 °C; please cross-check again.

L 138: Sacrificed is more likely a religious word; euthanize should be the appropriate wording.

L 185-186: The Duncan test should not be used for multiple means comparisons. Although it name suggests differently, this test does not take the repetition on tests into account that occur when you compared more than two means against each other. However, this must be done to correct the threshold of significance because a higher repetition of tests increases the probability to detect a false-positive result just by chance. Hence, this dataset should be revised applying more suitable post-hoc tests, e.g., Tukey's post-hoc test. Consult the following open-access source for more details. I also added the link where you can access the respective chapter on multiple comparisons within this e-book.

McDonald, J. H. 2014. Handbook of Biological Statistics, 3rd ed. Sparky House Publishing, Baltimore, MD, USA. (http://www.biostathandbook.com/multiplecomparisons.html).

L 193: use the abbreviation for Lactobacillus plantarum.

L 193: delete “probiotic”

Table 2: For Hb, the P value is not less than 0.05; thus, it should not be included in multiple comparisons.

Table 2: Table footnotes should be more descriptive; e.g., Data are presented as means with their pooled SEM; number of replicates, etc. It is recommended for other Tables also.

L 210: Change “greater” to “higher”.

L 215: delete superscript c; it is not available in the table.

L 254/258: Change “Imagine” to “Figure”

L 291/295: Change “Image” to “Figure”

L 300: Stope?

L 312-313: Please rephrase or rewrite this sentence for better clarification.

L 317: It is not necessary to cite Tables/Figures in the discussion; delete them throughout the text.

L 339: Change “greater” to “higher”.

L 340: Change “treatments” to “groups”.

L 363: Change “Some authors” to “Previous studies”.

L 367: Include the probiotic species names rather than using the general term "probiotics".

L 391-415: All bacteria names should be italicized.

References: Should be revised according to the journal format.

The manuscript contains significant errors in English grammar, sentence structure, and punctuation. Therefore, the paper needs careful editing by someone with expertise in technical editing paying particular attention to English grammar, technical expressions, spelling, and sentence structure.

Author Response

Dear Reviewer,

Thank you for your comments.  We inform you that all the corrections have been made and are highlighted in red in the text (corrected article).

R1. It is noted that the paper needs careful editing by someone with expertise in technical editing paying particular attention to English grammar, technical expressions, spelling, and sentence structure.

Answer 1. The manuscript was reviewed and corrected by a native scientist.

R2. L29: The TRT2 group.

Answer 2: Done

R3. L 30: Specify the name of the probiotic strain.

Answer 3: Done

R4. L 32: the CTRL and TRT1 groups showed.

Answer 4: Done

R5. L 43: By adopting.

Answer 5: Done

R6. L 50: use a comma after the word “thus”.

Answer 6: Done

R7. L 53-55: “feeding pigs diets supplemented” change to “diets supplemented”.

Answer 6: Done

R8. The hypothesis of this work should be included end of the introduction.

Answer 8. The hypothesis was added.

R9. L 83: Word capitalization should be consistent throughout the manuscript.

Answer 9: Done

R10. L 85: Abbreviation should be defined as its first appearance in the text; then, use abbreviation throughout the text.

Answer 9: Done

R11. L 93-96: What is the basis for dose selection? Please explain this and clearly mention it in the M&M.

Answer 11: The dose of the probiotic used is specified in materials and methods.

R12. Table 1: What is the difference between Soybean meal and Soymeal?

Answer 12:  One soybean has fat, and another is defatted, the terms in Table 1 are used.

R13: Table 1: Change “Grow Nucleus” to “Premix”

Answer 13: Done

R14. Table 1: Convert the unit “Kcal/kg” to “MJ/kg”

Answer 14: Done

R15. L 101-102: D3>> D3, B12>> B12

Answer 15: Done

R16. L 107-116: More details of housing conditions, e.g., temperature, humidity, ventilation, etc.

Answer 16. The details were indicated in materials and methods

R17. L 107-116: How about the growth performance and feed intake/refusal of piglets?

Answer 17: The details were indicated in the discussion.

R18. L 123: rpm should be converted into × g,

Answer 18. Done

R19. L 123: Generally, centrifugation to obtain plasma should be conducted at 4 °C; please cross-check again.

Answer 19: Done.

R20. L 138: Sacrificed is more likely a religious word; euthanize should be the appropriate wording.

Answer 20: Done

R21. L 185-186: The Duncan test should not be used for multiple means comparisons. Although it name suggests differently, this test does not take the repetition on tests into account that occur when you compared more than two means against each other. However, this must be done to correct the threshold of significance because a higher repetition of tests increases the probability to detect a false-positive result just by chance. Hence, this dataset should be revised applying more suitable post-hoc tests, e.g., Tukey's post-hoc test. Consult the following open-access source for more details. I also added the link where you can access the respective chapter on multiple comparisons within this e-book.

McDonald, J. H. 2014. Handbook of Biological Statistics, 3rd ed. Sparky House Publishing, Baltimore, MD, USA. (http://www.biostathandbook.com/multiplecomparisons.html).

Answer R21: We know that statistical analyzes always generate scientific contradictions. There are many methods of a posteriori comparison of means (see link). According to our experience when we use 3 or more experimental treatments and when the analysis of variance (for example) indicates statistical differences, the recommended a posteriori multiple comparisons of means is Duncan (1955), as it is the most discriminatory when the number of treatments is equal or less than 7 and is perhaps the most widely used post-hoc test in Animal Science. We are showing some articles published in high-impact journals that we use Duncan as a reliable method to determine the difference between means.

Betancur, C., Martínez, Y., Merino-Guzman, R., Hernandez-Velasco, X., Castillo, R., Rodríguez, R. & Téllez, G. 2020. Evaluation of oral administration of Lactobacillus plantarum CAM6 strain as an alternative to antibiotics in weaned pigs. Animals. 10(1218). http://doi.org/10.3390/ani10071218.

Betancur, C., Martínez, Y., Tellez, G., Avellaneda, C. & Velázquez-Martí, B. 2020. In vitro characterization of indigenous probiotic strains isolated from Colombian creole pigs. Animals. 10(1204). http://doi.org/10.3390/ani10071204.

Martínez Y., Orozco, E., Montellano, R. M., Valdivié, M. & Parrado, C. A. 2021. Use of achiote (Bixa orellana L.) seed powder as pigment of the egg yolk of laying hens. Journal of Applied Poultry Research. 30(2). https://doi.org/10.1016/j.japr.2021.100154.

Martínez, Y., Almendares, C., Hernández, C., Avellaneda, M. C., Urquía, M. N. & Valdivié, M. 2021. Effect of acetic acid and sodium bicarbonate supplemented to drink water on water quality, growth performance, organ weights, cecal traits and hematological parameters of young broilers. Animals. 11(7): 1865. https://doi.org/10.3390/ani11071865.

Martínez, Y., Altamirano, E., Ortega, V., Paz, P. & Valdivié, M. 2021. Effect of age on the immune and visceral organ weights and cecal traits in modern broilers. Animals. 11: 845. https://doi.org/10.3390/ani11030845.

Martínez, Y., Iser, M., Valdivié, M., Albarran, E. & Rosales, M. 2022. Dietary supplementation with Agave tequilana (Weber var. Blue) stem powder improves the performance and intestinal integrity of broiler rabbits. Animals. 12(9). DOI: https://doi.org/10.3390/ani12091117.

Melara, E.G., Avellaneda, M.C., Rondón, A.J., Rodríguez, M., Valdivié, M. & Martínez, Y. 2023. Characterization of autochthonous strains from the cecal content of creole roosters for a potential use as probiotics. Animals. 13: 455. DOI: https://doi.org/10.3390/ani13030455.

R22. L 193: use the abbreviation for Lactobacillus plantarum.

Answer 22. Done

R23. L 193: delete “probiotic”

Answer 23. Done

R24. Table 2: For Hb, the P value is not less than 0.05; thus, it should not be included in multiple comparisons.

Answer 24: The P value for the hemoglobin variable is 0.05, in addition, the multiple comparison of means (Duncan) indicated where there was a difference between the treatments. Below the results tables it was indicated that the statistical difference would be P≤0.05.

R25. Table 2: Table footnotes should be more descriptive; e.g., Data are presented as means with their pooled SEM; number of replicates, etc. It is recommended for other Tables also.

Answer 25: It was added number of replicates for all tables.

R26. L 210: Change “greater” to “higher”.

Answer 26: Done

R27. L 215: delete superscript c; it is not available in the table.

Answer 27: Done

R28. L 254/258: Change “Imagine” to “Figure”

Answer 2: Done

R29. L 291/295: Change “Image” to “Figure”

Answer 29: Done

R30. L 300: Stope?

Answer 30: The word was changed.

R31. L 312-313: Please rephrase or rewrite this sentence for better clarification.

Answer 31: The sentence was rephrased.

R32. L 317: It is not necessary to cite Tables/Figures in the discussion; delete them throughout the text.

Answer 32: Done

R33. L 339: Change “greater” to “higher”.

Answer 33: Done

R34. L 340: Change “treatments” to “groups”.

Answer 34. Done

R35. L 363: Change “Some authors” to “Previous studies”.

Answer 35: Done

R36. L 367: Include the probiotic species names rather than using the general term "probiotics".

Answer 36: The probiotic species was included.

R37. L 391-415: All bacteria names should be italicized.

Answer 37: Done

 R38. References: Should be revised according to the journal format.

Answer 38: Done

Reviewer 2 Report

Major comments

Introduction

-        L64-72: you should add more references about the administration of lactobacillus plantarum in pigs, such as 

·       Suo C, Yin Y, Wang X, Lou X, Song D, Wang X, Gu Q. Effects of lactobacillus plantarum ZJ316 on pig growth and pork quality. BMC Vet Res. 2012 Jun 25;8:89. doi: 10.1186/1746-6148-8-89.

·       Effect of dietary supplementation with Lactobacillus plantarum on growth performance, fecal score, fecal microbial counts, gas emission and nutrient digestibility in growing pigs. Animal Feed Science and Technology Volume 290, August 2022, 115295

·       Guerra-Ordaz AA, González-Ortiz G, La Ragione RM, Woodward MJ, Collins JW, Pérez JF, Martín-Orúe SM. Lactulose and Lactobacillus plantarum, a potential complementary synbiotic to control postweaning colibacillosis in piglets. Appl Environ Microbiol. 2014 Aug;80(16):4879-86. 

·       Shin D, Chang SY, Bogere P, Won K, Choi JY, Choi YJ, Lee HK, Hur J, Park BY, Kim Y, Heo J. Beneficial roles of probiotics on the modulation of gut microbiota and immune response in pigs. PLoS One. 2019 Aug 28;14(8):e0220843.

Materials and Methods

-        L91-96: report the commercial name of the products used in the trial

-         

Results 

-        Table 2: add a,b manuscripts in all hematological parameters

-        Table 3: add a,b manuscripts in all intestines (weight and length) traits

Minor comments

-        L60: … health and productivity of pigs..

-        L95: .. active compounds

-        L300: ..especially to stop or reduce..

-        L305:… variations in blood count

-        L332: .. have anti-inflammatory and antimicrobial effects,

-        L336: .. prolonged time modify iron homeostasis..

-        L406-415: use italics (Lactobacillus spp, Lactobacillus plantarumL. reuteriL. crispatus)  

-         

Author Response

Dear Reviewer,

Thank you for your comments. Honestly, your references were used and have been very valuable in improving our work.

We inform you that all the corrections have been made and are highlighted in red in the text (corrected manuscript).

Introduction

R1.     L64-72: you should add more references about the administration of lactobacillus plantarum in pigs, such as 

Suo C, Yin Y, Wang X, Lou X, Song D, Wang X, Gu Q. Effects of lactobacillus plantarum ZJ316 on pig growth and pork quality. BMC Vet Res. 2012 Jun 25;8:89. doi: 10.1186/1746-6148-8-89.

Effect of dietary supplementation with Lactobacillus plantarum on growth performance, fecal score, fecal microbial counts, gas emission and nutrient digestibility in growing pigs. Animal Feed Science and Technology Volume 290, August 2022, 115295Guerra-Ordaz AA, González-Ortiz G, La Ragione RM, Woodward MJ, Collins JW, Pérez JF, Martín-Orúe SM. Lactulose and Lactobacillus plantarum, a potential complementary synbiotic to control postweaning colibacillosis in piglets. Appl Environ Microbiol. 2014 Aug;80(16):4879-86. 

Shin D, Chang SY, Bogere P, Won K, Choi JY, Choi YJ, Lee HK, Hur J, Park BY, Kim Y, Heo J. Beneficial roles of probiotics on the modulation of gut microbiota and immune response in pigs. PLoS One. 2019 Aug 28;14(8):e0220843.

Answer 1: These references were used in the introduction.

Materials and Methods

R2.     L91-96: report the commercial name of the products used in the trial.

Answer 2: It was added to the commercial name.

Results 

R3.   Table 2: add a,b manuscripts in all hematological parameters.  Table 3: add a,b manuscripts in all intestines (weight and length) traits.

Answer 3: The superscript (a,b) was added to the parameters that were significant.

Minor comments

R4.  L60: … health and productivity of pigs.

Answer 4: the sentence was corrected.

R5.   L95: active compounds

Answer 5: the sentence was corrected.

R6.     L300: specially to stop or reduce.

Answer 6. The sentence was corrected.

R7. L305: variations in blood count  

Answer 7:  The sentence was corrected.

R8. L332: have anti-inflammatory and antimicrobial effects,

Answer 8: The sentence was corrected.

R9       L336: prolonged time modify iron homeostasis.

Answer 9: The sentence was corrected.

R10.     L406-415: use italics (Lactobacillus spp, Lactobacillus plantarumL. reuteriL. crispatus

Answer 10: Done.

Reviewer 3 Report

Comments to the Authors of manuscript number: animals-2405945 entitled “Effect of oral administration with Lactobacillus plantarum CAM6 on the hematological profile, relative weight of digestive organs and cecal traits in growing pigs”.

The Idea of the study could be interesting, however, it is not understood how it was possible to perform histological analysis on 4 animals. It is a simple tool, not expensive. All comments are below, and each comment should be taken into consideration.

1. L 15 - please avoid “we” in writing – it should be corrected

2. L 17 – do not use the form like “ in our works” – rephrase

3. L 26 – English

4. The clear hypothesis should be presented

5. The study design is poorly described, thus it is not known how long pigs received probiotics. Was possible to give to pig at the age of 138 days probiotics by the syringe?

6. how antibiotic was given? It should be described

7. How pigs were distributed within groups. How kept?

8. How piglets were divided into 3 groups? What were criteria?

9. L 119- how selected?

10. L 121 – what about hemolysis?

11. L 127 – this part should be rephrased. Monocytes are leukocytes

12. L 135 – these pigs were different than that subjected to blood sampling?

13. L 141- how the length was determine? Do the length depend on the force used?

14. L 147 – if 6 pigs were (L 136) killed, why only 4 were subjected to histomorphology? It is a methodological mistake. N=4 is too small. In this case the power test for each parameter should be given and it should be supported by the reference, where is information that these 4 pigs is enough.

15. N=6 were killed, N=4 for histology, and how n=3 for microbiology were chosen? It does not have any sense

16.  L 184- how uniformity was performed?

17. Table 2- use the SI units

18. Table 3 – weight should be given in g

19. there is a lack a description presenting the measurement of morphology, and what was exactly measured.

20. figure not image – how is known that there are exactly that bacteria – f

21. Figure 1 – why scale bares are different? Each image should be the same orientation and should include the same elements. It should be corrected.

22. L 339 – it is a poor indicator

23. There is no body weight data. It should be presented.

Author Response

Dear Reviewer,

Thank you for your comments. We inform you that all corrections have been made and are highlighted in red in the text (corrected manuscript).

R1. L 15. Please avoid “we” in writing – it should be corrected.

Comment 1: Done  

R2. L 17. Do not use the form like “in our works”–rephrase.

Comment 2: The paragraph line 17 was rearranged.

R3. L 26. English

Comment 3. A native Professor and researcher reviewed the English of the manuscript.

R4. The clear hypothesis should be presented.

Comment 4. The hypothesis was added.

R5. The study design is poorly described; thus, it is not known how long pigs received probiotics. Was possible to give to pig at the age of 138 days probiotics by the syringe?

Comment 5: The wording of the experimental design was improved and the probiotic biopreparation was used for 90 days. One of the aims of this work was to demonstrate that the strains of L. plantarum CAM6 had a positive response in pigs. We include in the recommendations to carry out a scale-up test with the future probiotic.

R6. How antibiotic was given? It should be described.

Comment 6: It was indicated in materials and methods.  

R7. How pigs were distributed within groups. How kept?

Comment 7: It was indicated in materials and methods.

R8. How piglets were divided into 3 groups? What were criteria?

Comment 8: The pigs were randomly distributed among the treatments.

R9. L. 119- how selected?

Comment 9. Pigs were randomly selected at the same time from 20 sows that gave birth on the same day. It was indicated in M & M.

R10. L 121 – what about hemolysis?

Comment 10: Tubes with ice packs were placed in airtight containers” (paragraph corrected). The normal erythrocyte count was achieved, and it was shown to have no effect on haemolytic activity.

R11. L 127. this part should be rephrased. Monocytes are leukocytes.

Comment 11: The paragraph was rearranged.

R12. L 135. These pigs were different than that subjected to blood sampling?

Comment 12: In the experiment, 12 pigs were used per treatment and hematology was performed on those same animals.

R13. L 141. How the length was determined? Do the length depend on the force used?

Comment 13: It was indicated in materials and methods.

R14. 14. L 147 – if 6 pigs were (L 136) killed, why only 4 were subjected to histomorphology? It is a methodological mistake. N=4 is too small. In this case the power test for each parameter should be given and it should be supported by the reference, where is information that these 4 pigs are enough.

Comment 14: This was a transcription error. 6 pigs were slaughtered for each treatment (2 per pen) for a total of 18 pigs.

R15. N=6 were killed, N=4 for histology, and how n=3 for microbiology were chosen? It does not have any sense

Comment 15: We used: n=6 were killed per treatment, n= 6 for histology and n=6 for microbiology.

R16.  L 184- how uniformity was performed?

Comment 16: As reported in the manuscript for the normality of the data the Kolmogorov Smirnov test was used and for the uniformity of the variance Bartlett's test was used, before carrying out the ANOVA, this clearly shows the data were not scattered and allowed to find significant differences among treatments.

R17. Table 2- use the SI units

Comment 17: The hematological parameters were changed to SI units (table 2)

R18. Table 3 – weight should be given in g.

Comment 18: Done

R19. There is a lack of a description presenting the measurement of morphology, and what was exactly measured.

Comment 19: It was indicated in materials and methods.

R20. figure not image – how is known that there are exactly that bacteria – f

Comment 20: The figure was deleted.

R21. Figure 1 – why scale bares are different? Each image should be the same orientation and should include the same elements. It should be corrected.

      Comment 21. Done.

R22-23. L 339 – it is a poor indicator. 23. There is no body weight data. It should be presented.

Comment 22-23: The final body weight was indicated.

Round 2

Reviewer 3 Report

I have no comments. Thank Authors for the response.